# Evolution of Interferon-Gamma Aptamer with Good Affinity and Analytical Utility by a Rational In Silico Base Mutagenesis Post-SELEX Strategy

**DOI:** 10.3390/molecules27175725

**Published:** 2022-09-05

**Authors:** Lianhui Zhao, Qionglin Wang, Yingai Yin, Yan Yang, Huifang Cui, Yiyang Dong

**Affiliations:** 1College of Life Science and Technology, Beijing University of Chemical Technology, Beijing 100029, China; 2Henan Key Laboratory of Children’s Genetics and Metabolic Diseases, Children’s Hospital Affiliated to Zhengzhou University, Zhengzhou 450018, China; 3College of Life Sciences, Zhengzhou University, Zhengzhou 450001, China

**Keywords:** aptamer, in silico, base mutagenesis, post-SELEX, interferon-gamma

## Abstract

The Systematic Evolution of Ligands by EXponential enrichment (SELEX) is conventionally an effective method to identify aptamers, which are oligonucleotide sequences with desired properties to recognize targets specifically and sensitively. However, there are some inherent limitations, e.g., the loss of potential high-affinity sequences during biased iterative PCR enrichment processes and the limited structural diversity of the initial library, which seriously restrict their real-world applications. To overcome these limitations, the in silico base mutagenesis post-SELEX strategy based on the low Gibbs free energy (ΔG) and genetic algorithm was developed for the optimization of the interferon-gamma aptamer (B1-4). In the process of evolution, new sequences were created and the aptamer candidates with low ΔG values and advanced structures were produced. After five rounds of selection, systematic studies revealed that the affinity of the newly developed evolutionary aptamer (M5-5) was roughly 10-fold higher than that of the parent aptamer (B1-4), and an aptasensor detection system with a limit-of-detection (LOD) value of 3.17 nM was established based on the evolutionary aptamer. The proposed approach provided an efficient strategy to improve the aptamer with low energy and a high binding ability, and the good analytical utility thereof.

## 1. Introduction

An aptamer, a kind of oligonucleotide sequence with a specific three-dimensional structure to selectively bind to a specific target, has many merits, such as a small size, customizable modification, long half-life, low cost, and low variability between batches. An aptamer is conventionally screened and isolated from the Systematic Evolution of Ligands by EXponential enrichment (SELEX) [1,2]. Although the SELEX is an effective screening method and many improvements have been made by researchers, it still has some inherent limitations. The structural diversity of oligonucleotide sequences in the initial library is practically limited. In addition, the biased iterative polymerase chain reaction (PCR) process in the SELEX makes it difficult to amplify sequences with advanced structures [3]. Therefore, the aptamers screened by traditional SELEX techniques are not always strong enough to detect targets.

Several post-SELEX strategies [4] have been proposed to make up for the shortcomings of the traditional SELEX, such as splitting or truncation, bivalent or multivalent construction, and chemical modification. Recently, the in silico post-SELEX strategy has emerged as a potential strategy to improve aptamers through an iterative process followed by in vitro experimental evaluation [5,6]. The computer-assisted method could efficiently optimize the aptamer through three main steps, including constructing the secondary and tertiary structures of the aptamer, predicting the structural conformation of the aptamer binding to the target, and optimizing the binding of the aptamer by mutation or modification. For instance, Song et al. engineered a high-affinity aptamer mutant that specifically binds to marine toxin gonyautoxin 1/4 [7]. Hsieh et al. successfully found one new sequence that could produce a relatively superior binding reaction with a prostate-specific antigen based on the known aptamers [8]. Wang et al. identified two aptamer mutants against the carcinoembryonic antigen exhibiting significantly high binding ability [9]. Thus far, the in silico post-SELEX strategy has been performed to engineer high-affinity aptamers for several targets. However, the thermodynamical stability of the aptamer has not been considered during the in silico post-SELEX program. Low ΔG was an important factor in the identification of an aptamer with high affinity [10,11]. Normally, the aptamer with the most thermodynamically stable structure has the lowest ΔG value [6,12,13]. In other words, as conformational flexibility is a key factor in limiting the affinity and specificity of aptamers, aptamers with a good binding ability mainly depend on stable conformation, especially in complex samples [14,15].

In the field of biomedical analysis, aptamers provide an alternative research platform for rapid detection and precise diagnosis [16]. Interferon-gamma (IFN-γ), an important cytokine, can act as an early indicator of infectious diseases, such as tuberculosis. Tuberculosis is one of the most catastrophic diseases and its transmission is very rapid [17,18]. Traditional antibody-based tuberculosis detection techniques commonly include the enzyme-linked immunosorbent assay (ELISA) and interferon-γ release assay (IGRAs) [19]. However, these methods have certain disadvantages, such as complicated operations and high costs. In contrast to antibodies, aptamers are still practically in a nascent stage without widespread use [20]. To further screen individuals efficiently for point-of-care testing, a model parent DNA aptamer (B1-4) against interferon-gamma was selected to optimize [21].

With the purpose of improving the performance of aptamers and promoting their application, as shown in Figure 1, an innovative post-SELEX screening method was proposed. The scheme mainly involved the following steps: (1) the generation of aptamer mutants from B1-4 using a lab-written program; (2) the analysis of the minimum free energy of a secondary structure to select the aptamer candidates using the batch calculation function of Mfold; and (3) the model simulation and prediction of the binding ability of the aptamer candidates for IFN-γ using a molecular docking program. In this way, all sequences were screened and generated in silico without bias, and the scheme was advantageous for efficiently exploring the evolutionary aptamer with low ΔG and high affinity. Furthermore, the affinity values of the aptamer mutant with the highest docking score and B1-4 were tested by biolayer interferometry (BLI), which is an optical technique to measure the surface–biomolecule interactions by analyzing the interference patterns [22]. Furthermore, a fluorescent assay based on the evolutionary aptamer for IFN-γ detection with high sensitivity and selectivity was developed.

## 2. Results and Discussion

### 2.1. Production of Evolutionary Aptamer against IFN-γ by In Silico Base Mutagenesis Post-SELEX

The in silico base mutagenesis post-SELEX process is an efficient method for the selection of an evolutionary aptamer against IFN-γ. Using an in silico base mutagenesis program and Mfold batch process, 10 mutants with low ΔG values were produced in each round, as shown in Appendix A. The secondary structure of B1-4 was predicted using the Mfold and its ΔG was −1.03 kcal/mol (Figure 1a). The docking score of the B1-4/IFN-γ complex was 1246.537 (Figure 1b). Furthermore, the key binding sites of B1-4 with IFN-γ were analyzed by a PLIP, and the results showed that the bases of G31, A32, C33, A34, T35, and C54 bind to the IFN-γ by hydrogen bonding (Appendix A). In the first round, the top 10 mutants (ΔG ranging from −3.18 to −4.76 kcal/mol) had lower ΔG than B1-4 and they were docked with IFN-γ, in which 4 mutants (M1-2, M1-5, M1-8, M1-9) with high ZDOCK scores indicated good docking results (Appendix A). For the four sequences, each was mutated to 250 sequences, which were used as the library for the next round of screening. In the second round, the ΔG values of the top 10 mutants ranged from −5.92 to −7.11 kcal/mol, in which four sequences (M2-6, M2-8, M2-9, and M2-10) with high ZDOCK scores were used for the third round of screening (Appendix A). After the programmed process, the ΔG values of the top 10 mutants were in the range of −8.44 to −9.44 kcal/mol in the third generation, −10.76 to −11.42 kcal/mol in the fourth generation, −12.46 to −14.13 kcal/mol in the fifth generation, and −14.57 to −16.29 kcal/mol in the sixth generation, indicating that each round of base mutagenesis could evolve sequences with lower ΔG than the previous round (Appendix A). In the fifth generation, the ΔG of the mutant M5-5 was −12.98 kcal/mol (Figure 1c), much lower than that of B1-4, and the ZDOCK score of 1543.334 was the highest score among all the mutants. The bases involved in the interactions of the M5-5/IFN-γ are mainly distributed in C2, G3, A9, T10, C11, C12, G14, and A20 (Appendix A). In the sixth generation, the ZDOCK score of the top mutant was 1406.824, much lower than that of M5-5, indicating that further mutation probably caused a worsening in the docking results. In addition, one aptamer mutant with positive ΔG (CCGCCCAAATCCCTAAGAGAAGACTATAATGACATCAAACCAGACACACTACACACGCA, 0.04 kcal/mol) was used as a negative control to be docked to IFN-γ. The ZDOCK score was 1124.301, much lower than that of B1-4. The result further indicated that low ΔG was an important factor for the aptamer binding IFN-γ. Thus, the mutant M5-5 with the highest ZDOCK score was determined to be the best aptamer candidate, binding the IFN-γ in all the mutants.

Furthermore, although there exist many bioinformatics tools for predicting secondary structure of aptamer sequences, further improvement in accuracy is still awaited. To demonstrate the change in the aptamer conformation in practice, the conformations of B1-4 and M5-5 before and after binding with IFN-γ in the PBS buffer were investigated by circular dichroism spectroscopy. As shown in Appendix A, the CD spectrum of B1-4 and M5-5 both have a negative peak around 245 nm and a positive peak around 275 nm, suggesting that they both have B-type DNA conformation. Obviously, the positive peak of M5-5 is higher than that of B1-4, and the negative peak is lower than that of B1-4, indicating that the evolutionary aptamer has more complementary base pairings and a folded structure, which is consistent with the predicted results. In addition, the peak intensity of the B1-4 circular dichroism suggested that its secondary structure also tends to form a folded structure, indicating that the aptamer with poor thermodynamic stability may not exist in the predicted form in practice, further indicating the necessity of optimization. Upon IFN-γ binding, the absorption intensity of the positive and the negative peaks has no significant change for B1-4 and M5-5. The results reveal that IFN-γ cannot induce the changes in the aptamers’ B-type conformation. Therefore, the structure of the evolutionary aptamer evolved in a way that was more favorable for binding to the target.

### 2.2. Generation of Evolutionary Aptamer with High Affinity

In order to verify the validity of the in silico screening result, the affinities of the evolutionary aptamer M5-5 and the parent aptamer B1-4 were characterized by BLI. As shown in Figure 2a,b, the BLI signal responses increased as the IFN-γ concentration increased from 125 to 2000 nM. The *K*_d_ values of M5-5 and B1-4 were 105.6 and 913.9 nM, respectively, which revealed that the affinity of M5-5 was roughly 10-fold higher than that of B1-4. In addition, as shown in Appendix A, the *K*_d_ values of the top mutants for each round of selection (M1-8, M2-6, M3-6, and M4-10) were determined to be 439.5, 481.6, 267.1, and 276.6 nM, respectively. The results showed that M5-5 has the best affinity for IFN-γ, which was consistent with the simulation results and indicated that the in silico base mutagenesis post-SELEX we proposed could be used to remarkably improve the aptamer binding ability.

### 2.3. Detection of IFN-γ Using Evolutionary Aptamer

In order to investigate the applicability of the evolutionary aptamer M5-5, a fluorescent assay for the detection of IFN-γ was constructed. As shown in Figure 3a, the molecular beacon (MB) was first designed to contain three parts, including a loop that hybridizes with the aptamer, a stem that contains five base pairs of a complementary sequence, and a fluorophore–quencher pair. Carboxyfluorescein (FAM) as a fluorophore was attached to one terminus of MB, and Black Hole Quencher 1 (BHQ1) was located at the other terminus. As shown in Figure 3b, in the absence of IFN-γ, the MB hybridized with the M5-5, resulting in the separation of the FAM and BHQ1 pairs and enhanced fluorescence. After introducing IFN-γ, due to the strong affinity of the aptamer with IFN-γ, the MB was released from the duplex of the M5-5/MB to form the terminal hybridization structure (i.e., the fluorophore and quencher were in close proximity). Thus, the fluorescence was efficiently quenched. To verify the successful hybridization of the M5-5/MB, as shown in Appendix A, the fluorescence emission spectra of the M5-5/MB duplex and MB were tested, respectively. The results showed that the fluorescence value of the M5-5/MB duplex recovered significantly.

To optimize the length of the MB, 250 nM M5-5 was mixed with MBs of different lengths (MB-10, MB-15, MB-20, and MB-25) at a molar ratio of 1:3. By measuring the fluorescence intensity of the M5-5/MB duplex before and after adding 250 nM IFN-γ, the fluorescence reduction rate was obtained. As shown in Figure 4a, the highest rate was obtained at MB-20. The fluorescence intensity of the M5-5/MB-10 duplex was weak owing to its short length. A longer loop in the MB sequence (MB-25) resulted in a lower fluorescence reduction rate. MB-25 had more nucleotides in the loop to promote the formation of the duplex with M5-5, thus limiting the M5-5 structure switching after the addition of the IFN-γ. Thus, the MB-20 was chosen for the IFN-γ analysis. Then, in order to ensure M5-5 fully hybridized with the MB, the ratio of M5-5 to MB was also investigated. In this experiment, the M5-5 at a final concentration of 250 nM was mixed with MB-20 at 1:1, 1:2, 1:3, and 1:4 molar ratios. As shown in Figure 4b, the fluorescence intensity increased with the increase in the molecular beacon concentration, indicating that more aptamers were bound with MBs, and few unbound aptamers were in the solution. When the concentration ratio was 1:3, the fluorescence intensity almost reached its maximum.

Under the optimal conditions, the responses of the proposed method based on the evolutionary aptamer M5-5 were investigated by varying the IFN-γ concentrations. As shown in Figure 4c,d, the value of (F_0_ − F)/F_0_ increased with an increasing concentration of IFN-γ and the fluorescence reduction rate performed a good linear correlation with the concentration of IFN-γ within a linear range of 1−250 nM. The calibration curve of (F_0_ − F)/F_0_ against the IFN-γ concentration was y = 0.106 C + 5.758 (R^2^ = 0.998), in which y was (F_0_ − F)/F_0_ and C was the concentration of IFN-γ. The limit of detection (LOD) of the aptasensor was defined as the concentration that corresponds to a three-fold standard deviation of the blank value of (F_0_ − F)/F_0_, calculated as 3.17 nM. This LOD is comparable to the previous method for IFN-γ detection based on fluorescence resonance energy transfer [23].

### 2.4. High Selectivity of the Assay Based on Evolutionary Aptamer

In order to determine the selectivity of this assay based on the evolutionary aptamer M5-5, the fluorescence intensity was measured in the presence of different interfering targets (Myoglobin (Mb), human serum albumin (HSA), and immunoglobulin G (IgG)) with a concentration of 250 nM under the same experiment condition as IFN-γ. As shown in Figure 5, only IFN-γ induced a significant fluorescence reduction, whereas less value was observed with the addition of other interferences, indicating that the assay based on the evolutionary aptamer exhibited high selectivity in the detection of IFN-γ. The three interfering proteins have also been used as interfering substances in the previous methods for the detection of IFN-γ and the results also confirmed that these methods had good selectivity [24,25].

## 3. Materials and Methods

### 3.1. Materials and Instruments

Human interferon-gamma (IFN-γ, greater than 95% purity) was purchased from SinoBio Biotech Ltd. (Shanghai, China). Human serum albumin (HSA), immunoglobulin G (IgG), and myoglobin (Mb) were purchased from Beijing Solarbio Science & Technology Co., Ltd. (Beijing, China). Phosphate-buffered saline (PBS, pH 7.4, 10×) was purchased from Beijing Kehua Jingwei scientific Co., Ltd. (Beijing, China). The synthesis and HPLC (high-performance liquid chromatography) purification of all oligonucleotides in this study were performed by Sangon Biotechnology Co., Ltd. (Shanghai, China). B1-4 (CCGCCCAAATCCCTAAGAGAAGACTGTAATGACATCAAACCAGACACACTACACACGCA) was screened by Cao et al. [21] and other sequences were listed in Appendix A.

The affinity of the aptamer was characterized by Octet RED96e System (Fortebio, CA, USA) (accessed on 10 July 2022). The fluorescent signal was measured by Synergy H1 Microplate Reader (Biotek, Winooski, VT, USA). The structure of aptamer was analyzed by J-815 spectropolarimeter (Jasco, TKY, Iapan).

### 3.2. Production of Aptamer Mutants by In Silico Mutagenesis

The B1-4, as a parent aptamer, was randomly mutated through an in silico base mutagenesis program designed by our team. The number of mutants in this study was about 1000 in each round, much higher than before [9]. The higher number of mutants means more possibilities of obtaining the sequence with better performance. Duplicate mutants were removed. Then, the secondary structures of aptamer mutants were predicted by the Mfold web server (http://www.unafold.org/mfold/applications/dna-folding-form.php) (accessed on 10 July 2022), based on a free-energy minimization algorithm and thermodynamic-based approach [26,27]. Using the shell script provided by Mfold, a batch process was written to fold secondary structures and arrange the sequences in ascending order of ΔG, generating a text file containing the sequences and secondary structures of dot–bracket information. The matched brackets represent base pairs, and dots represent unpaired bases. The top ten mutants were selected, and their three-dimensional (3D) structures were automatically modeled by RNA Composer (http://rnacomposer.ibch.poznan.pl/) (accessed on 10 July 2022) in batch mode [28,29]. Because the outcome was 3D RNA form, the modification was needed to ensure the structure of mutants in DNA coordinate. Molecular Operating Environment (MOE) DNA/RNA builder was used to convert the RNA to DNA, and energy minimization was used for determining low energy conformation. Finally, the prepared 3D DNA structures were docked with prepared IFN-γ (PDB: 1FG9) through the ZDOCK program (http://zdock.umassmed.edu) (accessed on 10 July 2022) [30,31]. The top four mutants with high docking scores were selected for the second cycle of base mutagenesis. The same steps of random mutagenesis and molecular docking were repeated 6 cycles. Finally, the aptamer mutant with the highest score was produced.

### 3.3. Model Simulation of Aptamer/IFN-γ

The prepared aptamer mutants and IFN-γ were docked using ZDOCK online server. The ZDOCK docking program, based on the Fast Fourier Transform, was an automated tool to search all possible binding modes between the aptamer mutants and IFN-γ, producing top 10 predictions with a high score. The best docking result of each mutant was selected to be compared with others. The interaction sites and interaction types of aptamer and IFN-γ were analyzed by automated protein–ligand interaction profiler server (PLIP, https://plip-tool.biotec.tu-dresden.de/plip-web/plip/index) (accessed on 10 July 2022) [32].

### 3.4. Biolayer Interferometry Assay

The equilibrium binding affinities (*K*_d_ values) of B1-4 and evolutionary aptamer were characterized by biolayer interferometry (BLI) through the following steps. First, the BLI chips coated with streptavidin (SA) were pretreated in baseline solution (Phosphate Buffered Saline, PBS, pH 7.4) for equilibration. Then, the BLI chips were immersed in solution containing biotin-modified aptamers with concentration of 500 nM to immobilize aptamers. Then, the chips functionalized with aptamer were immersed in PBS buffer for washing. Moreover, the IFN-γ in sample solution was captured by aptamers and the thickness of the biolayer increased. Finally, the chips were immersed in PBS solution for dissociation, and the thickness of the biolayer decreased. The reaction time of the above five steps was 120, 600, 180, 120, and 120 s, respectively. All steps were performed at 25 °C in a 96-well plate containing 200 μL solution in each well. The dissociation constant (*K*_d_) could be obtained by monitoring the biolayer thickness in real time.

### 3.5. Circular Dichroism Measurement

CD measurements were performed to investigate the conformations of B1-4 and M5-5 before and after binding with IFN-γ in the PBS buffer. The CD spectrum of aptamer (5 μM) or with IFN-γ (5 μM) was measured three times at a scanning rate of 200 nm/min in the range of 220–320 nm. The background signals of the PBS buffer were also measured and subtracted from the CD spectrum.

### 3.6. Fluorescence Detection of IFN-γ Using Evolutionary Aptamer

The complementary sequence as molecular beacon (MB) was designed to partially hybridize with evolutionary aptamer. Firstly, aptamer and MB were dissolved in PBS solution, respectively. Then, to allow the hybridization, the mixture of aptamer and MB was incubated at 95 °C for 5 min and cooled at 25 °C for 30 min. Then, different concentrations of IFN-γ were added to the mixture, thoroughly mixed, and then the obtained solutions were incubated at 25 °C for 30 min. Finally, 50 μL of the obtained solution was analyzed by fluorescence, with excitation wavelength at 480 nm and emission wavelength in the range of 510–600 nm. The fluorescence reduction rate was defined as y = (F_0_−F)/F_0_ × 100% (F_0_ and F are fluorescence intensity in the absence and presence of IFN-γ).

## 4. Conclusions

In this study, a novel in silico base mutagenesis post-SELEX strategy, based on minimum free energy and a genetic algorithm, was proposed to make up for the shortcomings of the traditional SELEX technology. The obtained evolutionary aptamer has a low ΔG, −12.98 kcal/mol, and a high ZDOCK score, 1543.334, indicating that it was conducive to binding with the IFN-γ. Moreover, it could also prove that the evolutionary aptamer had a good binding ability characterized by BLI. Then, a fluorescent assay was developed to detect IFN-γ by using an M5-5/MB duplex with an LOD as low as 3.17 nM. Our rational design of the in silico base mutagenesis post-SELEX strategy showed that the evolutionary approach was very promising for improving an aptamer, and the evolutionary aptamer combined with different technology could be further used to promote the application of early disease diagnosis and the prevention of disease transmission.

## Data Availability

Not applicable.

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
