# Peer review of "Evolution of Interferon-Gamma Aptamer with Good Affinity and Analytical Utility by a Rational In Silico Base Mutagenesis Post-SELEX Strategy"

_molecules, 2022, doi:10.3390/molecules27175725_

Round 1
Reviewer 1 Report
In the manuscript “Evolution of Interferon-gamma Aptamer for Better Affinity and Analytical Utility by a Rational in Silico Base Mutagenesis Post-SELEX Strategy”, Zhao et al. propose an in silico base mutagenesis post-SELEX strategy to improve the binding affinity of aptamers towards the interferon-gamma (IFN-γ), a protein that provides diagnostic information for tuberculosis. They selected a new modified version of B1-4 aptamer that, based on biolayer interferometry assays, experimentally presents a higher affinity towards IFN-γ with respect to the parent aptamer. Finally, they take advantage of their results to design a fluorescent assay for the detection of IFN-γ. The procedure adopted for the present study is suitable and the results obtained are interesting. However, the following aspects should be improved:
1) It is not clear why the in silico base mutagenesis post-SELEX has been stopped at the fifth round of selection. Is there a worsening in the deltaG values and ZDOCKs in the next cycles?
2) In order to understand the power of the presented method, one of the top mutants for each round of selection should be tested by BLI. These further experiments could confirm the correlation between the binding affinity and the calculated deltaG. One of the oligonucleotides with positive deltaG after the first round should be used as negative control.
3) At lines 107-114, it is not fully clear how the authors designed the docking experiments. They used different programs for different aptamers. Moreover, the docking results show that the two aptamers have different recognition mechanisms (see also lines 89-91 and 103-105). This seems quite strange for two 59-mer oligonucleotides that differ for only 5 residues (B1-4, 5’-CCGCCCAAATCCCTAAGAGAAGACTGTAATGACATCAAACCAGACACACTACACACGCA-3’; M5-5, 5’-CCGCCCAAATCCCGAAGGGAAGAGTGTAATGACGTCAAACCAGACACATTACACACGCA-3’). I believe that the structural prediction should be supported by stronger experimental evidence than only the docking results.
Otherwise, this prediction should be totally removed from the article.
4) At lines 166-168, the authors stated that “The limit of detection (LOD) of the aptasensor was defined as the concentration that corresponds to a three-fold standard deviation of the blank value of (F0-F)/F0, calculated as 3.17 nM, indicating a high detection sensitivity”. A comparison with other similar sensor used for the detection of IFN-γ should be reported.
5) The comparison with literature data should be extended also as concerns the selectivity of the assay described in paragraph 2.4.
Minor issues:
1) In the abstract, the sentence “In the process of evolution, new sequences were created and the aptamer candidates with advanced structure were produced with better binding affinity and stability” is not clear. In particular, the advancement in stability of produced aptamers is not experimentally verified.
2) Sometimes, “in silico” is not italicized.
3) At lines 107-108, in the sentence “In addition, we noticed that the second structure of M5-5 predicted
by Mfold was similar to B1-4 predicted by RNAfold” second should be substituted with secondary.
4) At lines 120-121, the sentence “the affinity of evolutionary aptamer M5-5 and parent aptamer B1-4 was characterized by BLI” should be replaced by “the affinities of evolutionary aptamer M5-5 and parent aptamer B1-4 were characterized by BLI”.
5) In all sections of the manuscript and also in the title, the comparative forms (better, higher, …) are wrongly utilized.
6) It could be helpful for the reader to move Figures S2 and S3 as a unique figure in the main text.
7) At lines 135-136, FAM and BHQ1 should be defined.
8) Some links reported in the manuscript do not work.
9) At line 210, “txt file” should be substituted with “text file”.
10) At lines 157-158, the sentence “indicating that more aptamers were bound with MBs and there
were few unbound aptamers in solution” should be substituted as follows: “indicating that more aptamers were bound with MBs and few unbound aptamers were in solution”.
11) At lines 247-250, the sentences “Then, to allow the hybridization, the mixture of aptamer and MB incubated at 95℃ for 5 min and cooled at 25℃ for 30 min. Then, different concentrations of IFN-γ were added to the mixture, thoroughly mixed and then incubated at 25℃ for 30 min” should be replaced as follows: “Then, to allow the hybridization, the mixture of aptamer and MB was incubated at 95℃ for 5 min and cooled at 25℃ for 30 min. Then, different concentrations of IFN-γ were added to the mixture, thoroughly mixed and then the obtained solutions were incubated at 25℃ for 30 min”.
12) At line 211 and in supplementary tables the Dot-bracket information should be explained.
13) The sentences at lines 44-53 is not fully clear. In particular, are all the sentences referred to in silico protocols? Maybe the authors could rewrite these sentences to clarify the concepts.
14) At lines 250-251, the sentence “Finally, 50 μL obtained solution was detected using fluorescence analysis, with excitation wavelength at 480 nm and emission wavelength in the range of 510–600 nm” should be replaced as follows: “Finally, 50 μL of the obtained solution was analysed by fluorescence, with excitation wavelength at 480 nm and emission wavelength in the range of 510–600 nm”.
15) At lines 227-228 in Materials and Methods section, the sentence “The best mutant was the evolutionary aptamer, which has the advantages of lower energy and better binding ability” should be removed as it has already been discussed in the results.
Reviewer 2 Report
The authors present an in silico aptamer evolution methodology and demonstrate it with the successful enhancement of an IFN-γ aptamer. The new leading aptamer and the original aptamer are characterised using BLI and an aptasensor is demonstrated with the new leading aptamer.
Comments:
-The literature review is very thin. The introduction should be expanded to fix this.
-English editing is required.
-A figure going over the sequence changes and the phylogeny of the M5-5 aptamer would be useful to the readers.
-Figure 2: Why is the angle of the docking image different? It should be the same angle for a good side to side comparison.
-Is the docking predicted binding very different between B1-4 and M5-5? This needs further analysis and discussion.
-There are a total of 5 point mutations between B1-4 and M5-5, yet your predicted structures are very different... Do you think this is accurate? You cant just put a sequence through M-fold and assume it is correct.
-Talk about possible future work and directions. Could you do a docking based negative selection to improve specificity?
Round 2
Reviewer 1 Report
The authors responded satisfactorily to all comments.
I only have further two minor revisions to suggest:
1. Although they satisfactorily answered question 5:
5) The comparison with literature data should be extended also as concerns the selectivity of the assay described in paragraph 2.4.
Response: Thank you for your precious comments and advice.
Revision: To evaluate the selectivity of the aptamer-based sensor, myoglobin, human serum albumin and immunoglobulin G have also been used as interfering substances in the following references. These results also confirmed the good selectivity of these methods for IFN-γ.
· Abnous, K.; Danesh, N.M.; Ramezani, M.; Alibolandi, M.; Hassanabad, K.Y.; Emrani, A.S.; Bahreyni. A.; Taghdisi, S.M.; A triple-helix molecular switch-based electrochemical aptasensor for interferon-gamma using a gold electrode and Methylene Blue as a redox probe. Microchim. Acta. 2017, 184, 4151–4157.
· Jin, H.; Gui, R.J.; Gao, X.H.; Sun, Y.J.; An amplified label-free electrochemical aptasensor of γ-interferon based on target-induced DNA strand transform of hairpin-to-linear conformation enabling simultaneous capture of redox probe and target. Biosens. Bioelectron. 2019, 145, 111732.
No revision was made to the manuscript. I suggest to properly cite the indicated papers.
2. The title in the Supplementary Material file has not been updated.
Reviewer 2 Report
"Response: Thank you for your precious comments. The 5 point mutations result in M5-5 having more complementary base pairings and it is easy to form stem-loop structures. The mfold server predicts the secondary structures of aptamers based on free-energy minimization algorithm and thermodynamic-based approach. In addition, the Mfold web server is the most widely applied bioinformatic tool to determine the secondary structures of aptamers. Then, the conformations of B1- 4 and M5-5 were investigated by circular dichroism spectroscopy."
Mfold is computational structure prediction and is frequently wrong. Just Mfold the sequence of any aptamer that has its crystal structure solved and you will see. Your 5 point mutations almost certainly did not result in a completely new folding that just so happened to bind to the same protein target. Looking at your figure S1, do you honestly think B1-4 is held together by 3 base pairs? If you look at the original paper you got the B1-4 aptamer from you will see their predicted structure is very similar to you new M5-5 aptamer... Redo your predicted structures and perhaps the retry the docking.
